# Diagnostic Accuracy of Serum Hyaluronan for Detecting HCV Infection and Liver Fibrosis in Asymptomatic Blood Donors

**DOI:** 10.3390/molecules26133892

**Published:** 2021-06-25

**Authors:** Itatiana F. Rodart, Madalena M. Pares, Aline Mendes, Camila M. Accardo, João R. M. Martins, Cleidenice B. Silva, Fabrício O. Carvalho, José A. Barreto, Mitermayer G. Reis, Ivarne L. S. Tersariol, Helena B. Nader

**Affiliations:** 1Departamento de Bioquímica, Escola Paulista de Medicina, Universidade Federal de São Paulo (UNIFESP), São Paulo 04021-001, Brazil; irodart@yahoo.com.br (I.F.R.); alina.mendess@gmail.com (A.M.); camila.accardo@gmail.com (C.M.A.); jrmacmartins@yahoo.com.br (J.R.M.M.); 2Associação Beneficente de Coleta de Sangue (COLSAN), São Paulo 04038-000, Brazil; madalena.pares@gmail.com (M.M.P.); barbosa_nice@ig.com.br (C.B.S.); f.carvalho@hc.fm.usp.br (F.O.C.); jabarreto5660@gmail.com (J.A.B.); 3Laboratório de Hepatites Virais, Centro de Pesquisas Gonçalo Muniz, Fundação Oswaldo Cruz, Salvador 40296-710, Brazil; mitergreis@gmail.com

**Keywords:** serum hyaluronan, diagnostic accuracy, liver fibrosis, liver enzymes, HCV infection, blood donors

## Abstract

*Background*: The disease caused by hepatitis C virus (HCV) is asymptomatic, silent, and progressive liver disease. In HCV-infected patients the increase in serum HA is associated with the development of hepatic fibrosis and disease progression. *Methods*: HCV-RNA detection was performed in all serological samples of blood donors that tested positive using HCV Ultra ELISA. Determination of hyaluronan (HA) was performed in positive HCV samples using ELISA-like fluorometric method. The HA content was compared to HCV viral load, genotype of the virus, liver fibrosis as well as ALT and GGT liver biomarkers. *Results*: Persistently normal ALT (<40 U/L) and GGT (<50 U/L) serum levels were detected in 75% and 69% of the HCV-Infected blood donors, respectively. Based on ROC analysis, the HA value < 34.2 ng/mL is an optimal cut-off point to exclude HCV viremia (specificity = 91%, NPV = 99%). Applying HA value ≥34.2 ng/mL significant liver fibrosis (≥F2) can be estimated in 46% of the HCV-infected blood donors. HA serum level (≥34.2 ng/mL) associated with a high ALT level (>40 U/mL) can correctly identify HCV infection and probable liver fibrosis (sensitivity = 96% and specificity = 90%) in asymptomatic blood donors. *Conclusions*: A high level of HA (≥34.2 ng/mL) in association with ALT (≥40 U/L) in serum can provide a good clinical opportunity to detect HCV-infected asymptomatic persons that potentially require a liver biopsy confirmation and antiviral treatment to prevent the development of advanced liver fibrosis or cirrhosis.

## 1. Introduction

The disease caused by the hepatitis C virus (HCV) is asymptomatic, silent, and progressive disease, characterized by liver injury, inflammation, and fibrosis that can lead to the development of hepatocellular carcinoma [1]. The fibrosis resulting from liver inflammation is characterized by the deposition of extracellular matrix (ECM) components such as collagen [2]. The development of fibrosis changes the hepatic architecture and the hepatocellular function and causes irregularities in the hepatic microcirculation [3].

The gold standard test used to assess the degree of liver fibrosis is a liver biopsy, which is an invasive procedure that, in some cases, may even lead to death due to sampling error [4]. For these reasons, non-invasive methods are desirable. Some studies have used serum markers to diagnose the degree of fibrosis and monitor its progression; others show that the combination of serological markers of hepatic fibrogenesis is efficient to evaluate liver histology [5,6,7,8]. The APRI method—aspartate aminotransferase (AST) and the odds ratio of platelets—considers the ability of serum AST and the platelet count to predict the degree of hepatic fibrosis [9]. Continued serum measurements of AST, alanine aminotransferase (ALT) and γ-glutamyl transferase (GGT) have been used to monitor patients [10,11,12]. In addition to these markers, some ECM components, such as procollagen type III, laminin, and hyaluronic acid (HA), have been used as biomarkers to predict liver fibrosis [13,14,15]. In HCV patients, the increase in serum HA is associated with the development of hepatic fibrosis and disease progression [16]. The HA serum concentration increases with the progression of the liver disease [17]. HA is a glycosaminoglycan synthesized by mesenchymal cells and degraded by hepatic sinusoidal cells; specific receptors mediate these processes [18].

We have postulated that markers of extracellular matrix (ECM) metabolism, such as hyaluronan levels, may have an additional prognostic value, as they might reflect not only the stage of liver disease but also the rate of ECM turnover and, hence, the rate of disease progression. HA is particularly interesting because it is enriched in matrices undergoing remodeling processes such as morphogenesis, wound repair, and tumor growth [19]. HA regulates cell migration, proliferation and differentiation and activates intracellular signaling cascades in tissue injury and repair [20]. HA levels in serum are elevated during liver regeneration after hepatectomy in humans, strongly suggesting that HA may be a marker of liver repair diseases [21]. HA signaling in tissue regeneration seems to be part of a conserved response to tissue injury. Recently, hyaluronic acid has been directly involved in the development of liver fibrosis, with hyaluronan synthase 2 (HAS2) having a central role in the production of HA, activation of hepatic stellate cells (HSCs) and in the progression of liver fibrosis. HA promotes HSC activation and liver fibrosis through Notch1 signaling. In mice lacking *Has2*, both HA and liver fibrosis levels were reduced; conversely, mice overexpressing *Has2* overproduced HA, promoting HSC activation and exacerbating liver fibrosis [22].

In the present work, the content of hyaluronan in serum was investigated in asymptomatic HCV-infected blood donors. The HA content was compared to the HCV viral load, the genotype of the virus, ALT and GGT, which are considered biomarker of liver function. Our main purpose here is not to create a method to test HCV-infected blood donors instead of HCV PCR, but rather to provide input to validate the serum HA detection to identify asymptomatic HCV-infected patients who are developing liver disease silently. Here, the present data give a possibility to non-invasively reveal liver fibrosis and initiate medical investigation and if needed treat blood donors found to be HCV RNA positive before they develop major liver disease.

## 2. Results

### 2.1. HCV ELISA, Genotype, and Viral Load in Blood Donors Samples

The presence of RNA-HCV was detected in 99 blood donors, all of them OD > 16. HCV genotype 1 (82.8%) is the most prevalent genotype found in the blood donors group, followed by type 3 (13.1%) (Appendix A). We were not able to identify the viral genotype in 4.1% of RNA-HCV positive samples. By Real-Time PCR the mean HCV viral load was (4.07 ± 0.14) × 10^3^ copies/mL. No significant differences were found between the viral load and the different genotypes: (4.2 ± 0.2) × 10^3^ copies/mL for genotype 1 and (3.7 ± 0.3) × 10^3^ copies/mL for genotype 3 (Appendix A).

### 2.2. Liver Biomarkers ALT, GGT, and HA in Blood Donors Infected with HCV

Because HCV infection can promote asymptomatic liver injury and fibrosis, we decided to investigate the relationship between viral infection and the liver biomarkers HA, ALT and GGT in blood donors. A strong significant difference (*p* < 0.001) was observed for the values for ALT between the HCV-RNA positive and healthy blood donors (Figure 1A). A significant difference (*p* < 0.0276) in GGT was found between the HCV-RNA positive and negative samples (Figure 1B). Additionally, a strong significant difference (*p* < 0.001) was observed in the HA serum levels between the HCV-RNA negative and HCV-RNA positive samples (Figure 1C). No significant differences were observed in the HA content according to the viral genotype (genotype 1, 40 ± 4 ng/mL; genotype 3, 35 ± 11 ng/mL, *p* = 0.81). These results clearly show that the HA, ALT and GGT serum levels are increased in blood donors infected with HCV, strongly suggesting that most blood donors infected with HCV are developing asymptomatic liver injury and liver fibrosis. However, ALT serum levels were persistently normal (<40 U/L) in 75% of the HCV-infected blood donors and only elevated in 25% HCV-infected blood donors, GGT levels were also persistently normal (<50 U/L) in 71% of the HCV-infected blood donors and only elevated in 29% of the HCV-infected blood donors. These results indicate the need to perform an additional test to reveal cases of liver disease due to HCV infection.

### 2.3. Hyaluronic Acid Serum Level Is Associated with Antibody Titers Anti-HCV

We examined the anti-HCV serum levels from blood donor samples as a method for detecting HCV infection and compared those data to the gold standard HCV RNA test (Table 1). ELISA assays are shown in Table 1 with respect to the different HCV antibody titer levels (optical density, OD), low antibody titer (OD < 16) and high antibody titer (OD ≥ 16). We observed that 96 of the 99 blood donors infected with HCV were correctly identified using the second trial anti-HCV antibody values of OD ≥ 16 (sensitivity = 97%). As a screening test, a negative result (OD < 16) provided excellent evidence for the absence of HCV infection in blood donors (NPV = 98%).

Serum HA is a clinically useful non-invasive biomarker for liver fibrosis in HCV-infected patients. The distributions of the HA content among the blood donors that gave positive results in the trial anti-HCV ELISA assay and in the additional ELISA assays are shown in Figure 2 with respect to the different HCV antibody titer levels, low antibody titer (OD < 16) and high antibody titer (OD ≥ 16). A strong significant difference (*p* < 0.0001) between the values for HA was observed when comparing the ELISA results for low antibody titer, OD < 16 and high antibody titer, OD ≥ 16. Additionally, no significant differences (*p* = 0.53) were observed in the HA serum levels between healthy blood donors and the low HCV antibody titer group, OD < 16. More, no significant differences (*p* = 0.1179) were observed in the HA serum levels between the high HCV antibody titer group, OD > 16, and HCV-RNA positive blood donors.

Previously, we demonstrated in a cohort of 206 biopsy patients with chronic hepatitis C that HA >34.2 ng/mL can predict, with high certainty (AUROC 88%), significant liver fibrosis (≥F2) according to the METAVIR scoring system [17]. Therefore, the HA cut-off value greater that 34.2 ng/mL was chosen for identifying presence of significant liver fibrosis in blood donors. Applying HA value greater that 34.2 ng/mL, the presence of significant fibrosis (F2–F4) can be predicted in 44 (46%) of the 96 HCV RNA–positive asymptomatic blood donors, with high certainty (AUROC = 0.88). Liver cirrhosis can be estimated in 11 (12%) out of 96 HCV-infected blood donors, predicted by HA value greater that 78.6 ng/mL, with 91% certainty [17]. Also, initial fibrosis (F0–F1) was predicted in 52 (54%) of the 96 HCV RNA–positive asymptomatic blood donors. Taken together, the present data show that 54% of the asymptomatic HCV-infected blood donors have initial fibrosis (F0–F1), 34% have moderate/severe fibrosis (F2–F3) and 12% have liver cirrhosis.

Interesting, 20 (45.5%) of the 44 HCV-infected blood donors with normal ALT levels also had significant liver fibrosis (F2–F4), predicted by HA value greater that 34.2 ng/mL. None with normal ALT levels (<40 U/L) had liver cirrhosis and all HCV-infected blood donors with elevated ALT levels (25.6%) also had significant liver fibrosis predicted by HA value greater that 34.2 ng/mL.

### 2.4. Correlation between HA and Liver Enzymes in HCV-Infected Blood Donors

It was evaluated the Spearman’s correlation coefficients between the content of hyaluronic acid and ALT in serum of HCV-infected blood donors, N = 88 (Figure 3A) or GGT, N = 88 (Figure 3B). A strong significant correlation was between HA and ALT serum level (ρ = 0.848) and a moderate significant correlation was found between HA and GGT serum level (ρ = 0.442). No correlation was observed between the serum levels of ALT (Figure 3C) and GGT (Figure 3D) with HA in the healthy blood donors. Also, the percentage of HCV-infected blood donors with significant liver fibrosis increased with increasing ALT and GGT levels (data not shown).

### 2.5. Analysis of Diagnostic Parameters of HA, ALT and GGT by ROC Curves

The receiver operator characteristic (ROC) curves of the different non-invasive liver biomarkers for the diagnosis of HCV infection are represented in Figure 4. For the discrimination of HCV viremia GGT, ALT and HA the areas under ROC curves (AUROC) were (Mean ± SE), 0.67 ± 0.04, 0.75 ± 0.03, and 0.82 ± 0.03, respectively. It was observed that HA showed the best diagnostic performance to discriminate HCV infection in blood donors. Based on ROC analysis, we choose the HA value of <34.2 ng/mL as the optimal cutoff point to exclude HCV viremia in asymptomatic blood donors (specificity = 91% and NPV = 100%). Taking in consideration this cutoff value, HA level of 34.2 ng/mL or more also offered a discriminatory ability to identified significant liver fibrosis (≥F2) in 46% of the HCV-infected asymptomatic blood donors [17].

It has been demonstrated that elevated serum ALT levels (above 40 U/L) can identify liver injury [23]. ALT is elevated in most patients with acute or chronic liver disease [24]. Using ALT values above the higher cut-off level (>40 U/L) as a gold standard test to confirm hepatic lesion in HCV infection and a high HA level (>34.2 ng/mL) as a comparative test to detect HCV infection and probable liver fibrosis in blood donors (Table 2), we observed that the HA screening test developed in this study with a cut-off of ≥34.2 ng/mL correctly identified 24 of the 25 blood donors infected with HCV (sensitivity = 96%). As a screening test, a negative result (ALT < 40 U/L and HA < 34.2 ng/mL) provided an excellent marker for the absence of HCV infection in blood donors (NPV = 99.5%) and, in this initial analysis, correctly identified 184 of the 204 blood donors who were not infected with HCV (specificity = 90%). These results indicate that a high HA serum level (≥34.2 ng/mL) associated with a high ALT level (>40 U/mL) can correctly identify HCV infection and probable liver fibrosis (sensitivity = 96% and specificity = 90%) in asymptomatic liver injury caused by HCV infection in blood donors.

## 3. Discussion

Chronic HCV infection constitutes a global health and economic problem [25] and is the leading cause of chronic liver disease in the world. This chronic infection predisposes the patient to the development of cirrhosis and hepatocellular carcinoma with life-threatening complications [26,27]. Because most HCV-infected persons are asymptomatic and are unlikely to be aware that they are infected, the general clinical consensus is that an improvement in current HCV testing practices is necessary to identify HCV-infected people that do not present an overt clinical picture of liver disease [27].

It has been shown that most asymptomatic HCV-infected patients are detected when they donate blood [28], and viral replication is only present in a variable proportion. The evidence of viremia as measured by the presence of HCV-RNA defines the diagnosis of ongoing HCV infection, independent of the liver histology findings [29]. Recently, it has been shown that the detection of high antibody levels (OD ≥ 20) by third-generation anti-HCV tests is an excellent predictor of viremia [27]. Our data complement this finding, identifying cases of HCV infection characterized by HCV RNA positive test and low levels of anti-HCV antibodies (OD < 16, Table 1). Despite of HCV RNA test needs to be done in blood banks as the only reliable to detected HCV-infection, our data indicates that HA test can complement testing of blood banks for HCV that use anti-HCV antibody tests, as it can reveal viremic donors negative for anti-HCV antibodies.

There was a 0.33% prevalence of anti-HCV positive blood donors among those included in this study. These results are like those found in other studies performed in Brazilian blood banks in which the prevalence of anti-HCV ranges from 0.38–2.6% [30,31]. Also, the prevalence of HCV in the United States is 1.6% and is 0.2% in the UK [32,33,34].

The genotype distribution found in this study did not differ from the existing data in Brazilian studies. Previous studies have shown that HCV genotype 1 is most prevalent (67.9%) in the samples of blood donors from the mid-west and southeast regions, followed by genotypes 3 (29.1%) and 2 (3%) [35,36]. We obtained similar results in our study, but we did not detect genotype 2 (Appendix A).

We observed a 2.3-fold increase in GGT in blood samples from HCV-RNA positive donors and a 3.5-fold increase in ALT (Figure 1). These results are similar to data found in the literature, with HCV-RNA positive patients [37,38]. According to the literature, 7.5–46% of patients with serum HCV-RNA have normal ALT serum levels [39], a significant progression of liver fibrosis in approximately 20–30% of patients with normal ALT levels, and the development of hepatocellular carcinoma has been described in many cases with persistently normal ALT [40,41]. Here, we are showing that 20 (45%) out of 44 HCV-infected asymptomatic blood donors with significant liver fibrosis (HA ≥ 34.2 ng/mL) also have normal ALT levels (<40 U/L), and this observation might have major implications for performing liver biopsies in HCV-infected persons with normal ALT serum levels. Indeed, ALT serum levels are not very indicative for slow processes in the liver like chronic hepatitis C fibrosis/cirrhosis [42]. However, elevated ALT levels are associated with an increased risk of liver-specific mortality [23]. Indeed, in the USA population, high serum level of ALT (above 40 U/L) has been recently used to discriminate subjects infected with HCV from those at low risk of liver disease [43].

The assessment of the presence and severity of liver fibrosis is critical for the determination of HCV treatment strategies. Patients with absent or minimal fibrosis are treated less aggressively and often wait for an appropriate treatment regime. Patients with advanced liver fibrosis and cirrhosis require considerable attention, including frequent biochemical and imaging tests for the evaluation of complications or carcinomas [44].

It has been shown that hyaluronic acid in serum can be utilized as a reliable surrogate marker to distinguish the three clinical stages of liver fibrosis: absent/minimal (Ishak stages 0–1), intermediate (Ishak stages 2–3), and advanced/cirrhosis (Ishak stages 4–6) [45,46]. Our data shows that as a screening test, an HA level of <34.2 ng/mL correctly identified 91% of those blood donors without HCV infection. Also, low HA level (<34.2 ng/mL) is a high negative predictive value (100% certainty) for identifying the absence of HCV infection and clearly divides viremic from non-viremic blood donors.

McHutchison et al. showed that HA has the greatest clinical utility for the exclusion of patients with significant fibrosis and cirrhosis, as a serum level of HA *<* 60 ng/mL has negative predictive values of 99% for cirrhosis and 93% for advanced hepatic fibrosis [42]. HA is also a highly predictive marker of liver-related death in HCV-infected individuals [47]. In HCV-coinfected persons with HIV, the risk of liver-related death increased 5-fold for patients with HA > 75 ng/mL and 30-fold for patients with HA > 250 ng/mL, compared to patients with normal HA [48]. Combining liver stiffness by FibroScan with HA ≥ 200 ng/mL provides superior prognostic performance to detected liver-related mortality in chronic hepatitis C [49].

Our results indicate (Table 2) that an HA level ≥ 34.2 ng/mL associated with ALT > 40 UI/L correctly identified HCV infection (sensitivity = 96% and specificity = 90%) and could potentially identify liver fibrosis with high certainty (AUROC = 0.88) in blood donors with asymptomatic HCV infection. Therefore, it is likely that the strong correlation observed between HA level and ALT serum levels (Figure 3A) represents an indirect mode of expressing the functional correlation between liver fibrosis and the degree of liver injury as determined by the ALT serum level, a primary screening parameter for liver diseases.

The present data show that asymptomatic blood donors that had high anti-HCV antibody titers also exhibited high HA serum levels (Figure 2). The serum level of HA is significantly (*p* < 0.001) associated with HCV viremia (Figure 1C) in asymptomatic blood donors. HA level also correlates with ALT and GGT liver enzymes in serum of the HCV-infected blood donors (Figure 3A,B). These results clearly indicate that HA can be used as a sensitive marker of active hepatic injury and fibrosis triggered by HCV infection. These results clearly indicate that HA and ALT can be used as markers of active hepatic injury triggered by HCV infection.

In patients with developing fibrotic liver disease HA serum levels increase. The pathogenesis of hepatic fibrosis can be divided into two stages capable of promoting the accumulation of HA in the serum. In the initial stage of the disease, the expression of hepatic HA synthase 2 (HAS2) is markedly up regulated by transforming growth factor–β in activated hepatic stellate cells (HSCs) and in murine and human liver fibrosis [22]. Moreover, increased HA promotes fibrogenic, proliferative, and invasive phenotypes of HSCs through activation of CD44 and TLR4 in an autocrine manner [22]. Later as severe liver fibrosis and cirrhosis are established impairment of the sinusoidal endothelial cells, which internalize and remove HA from circulation, results in even greater elevation of serum HA [50]. Therefore, the correlation of serum HA and liver fibrosis make HA a valuable biomarker for chronic liver disease. Indeed, HA serum level of <34.2 ng/mL discard significant liver fibrosis (≥F2) in blood-donors. Taken together, the use of HA as a non-invasive biomarker is a powerful and cost-effective way to exclude significant liver fibrosis and cirrhosis in the asymptomatic HCV-infected blood donors.

## 4. Materials and Methods

### 4.1. Serum Samples

Serum samples from Blood Collection Benefit Association (COLSAN) blood donors (N = 50,607) were collected in São Paulo (SP, Brazil), and were serologically tested for hepatitis B and C viruses, HIV I and II, HTLV 1 and 2, syphilis and Chagas disease. Those samples that were only positive for anti-HCV antibodies were aliquoted and stored at −70 °C until use. Samples from age- and sex-matched healthy donors with negative serological results were used as controls. All blood donors were formally informed of the project and signed a consent form. This study was approved by the Ethics Committee of the Federal University of São Paulo–Escola Paulista de Medicina according to Brazilian law number 1297/05.

### 4.2. HCV-ELISA Tests

Initially, the samples were assayed by the Hepanostika^®^ HCV Ultra BioMérieux (Beijing United Biomedical Co. Ltd., Beijing, China), a 3rd-generation ELISA test that detects antigen epitopes C22p, C200 and NS5r. Samples that tested positive were then subjected to a second, more sensitive anti-HCV ELISA using the Ortho^®^ HCV 3.0 ELISA Test System Enhanced SAVe (Ortho Clinical Diagnostic, Piscataway, NJ, USA), both following the manufacturer’s guidelines. The second ELISA test provided results of optical density (OD) that can be separated into groups with OD < 16 or OD > 16. Among the 50,607 blood donor samples, 308 tested positives using the first trial anti-HCV ELISA, and 168 tested positives using the second anti-HCV assay (43 tested positive with OD < 16 and 125 tested positive with OD ≥ 16) with a total prevalence of 54.55% of repeatedly reactive samples (168/308).

### 4.3. HCV-RNA Assays

HCV-RNA was extracted from all serological samples that tested positive using the Hepanostika^®^ HCV Ultra ELISA (N = 308) with 250 μL of serum and Trizol^®^ LS reagent (Invitrogen LifeTechnologies, Carlsbad, CA, USA) according to the manufacturer’s instructions. The extracted HCV-RNA was immediately transcribed into cDNA using random primers (Prodimol, Madison, WI, USA), and the cDNA was subjected to nested PCR using specific primers (numbers 939-209 and 940-211, LGC Biotecnologia (São Paulo, Brazil) that cover the 5′-untranslated virus, as described [51]. The second PCR product was subjected to 1.5% agarose gel electrophoresis using Tris-acetate-EDTA buffer and ethidium bromide, and the 251 bp fragment was detected by ultraviolet light.

The positive HCV-RNA samples were genotyped by RFLP as described [52]. The digestion reaction was proceeded for 4–16 h after adjustment with enzyme reaction buffer. The reactions were incubated at 37 °C in the presence of 10 units of each enzyme: (a) RsaI-HaeIII (Invitrogen LifeTechnologies) and (b) HinfI (Invitrogen LifeTechnologies)—MvaI (New England Biolabs, Ipswich, MA, USA). The digestion products were subjected to 4% MetaPhor agarose (BMA, Rockland, WA, USA) gel electrophoresis in Tris-acetate-EDTA and ethidium bromide, and the digestion fragments were visualized under UV light. The combination of the reaction band patterns of the two enzymes was used to deduce the HCV genotypes [53]. The HCV-RNA samples that were undetected in a first extraction were subjected to a second extraction for confirmation.

The HCV viral loads in the HCV-RNA positive samples were determined using the cDNA of the positive nested PCR samples by a real-time PCR reaction. Primers, a 5′-NCR region probe and TaqMan^®^ Universal Master Mix (Applied Biosystems, Foster City, CA, USA) were combined to give a final volume of 50 μL, and the 40 PCR reaction cycles were performed as follows: 2 min, 50 °C; 10 min, 94 °C; 30 s, 94 °C; 1 min, 60 °C [54]. The amplified products were analyzed with the ABI PRISM 7500 SDS (Applied Biosystems).

### 4.4. Liver Biomarks ALT, GGT and HA Assays

The ALT and GGT biochemical measurements were performed using an AU640 Chemistry-Immuno Analyzer (Olympus Corporation, Tokyo, Japan). The level of HA was determined in all anti-HCV positive serological samples using a highly specific, extremely sensitive ELISA-like fluorimetric method that employed a biotinylated HA-binding protein and europium-labeled streptavidin [55]. This methodology detects HA concentrations as low as 0.2 ng/mL. The samples were assayed in triplicate.

The analysis of the different parameters, namely: AH, ALT, GGT, HCV-RNA test and second trial, anti-HCV ELISA were done all 308 blood samples that tested for first trial anti-HCV ELISA (Hepanostika^®^ HCV Ultra BioMérieux). Of the 308 samples previously selected by the first ELISA trial, 99 blood donors tested positive for the presence of viral RNA and 209 tested negatives. The 209 blood donors that were false positive in the first ELISA screening represent the population of donors not infected with HCV; and the 99 donors who were true positive for the presence of viral RNA comprise the donor population infected with HCV. However, some serum samples from blood donors were lost while performing the tests because they have low volume of serum that prevents the analysis of any parameter. This loss was random, not involving any bias in the selection of samples that should be analyzed.

### 4.5. Determination of Liver Fibrosis

Determination of liver fibrosis in blood donors was performed according to the level of serum HA concentration. Serum HA was the parameter that alone presented the best diagnostic accuracy in the assessment of hepatic fibrosis in a total of 206 hepatitis C virus RNA-positive biopsied patients [17]. HA levels strongly correlated with liver disease stage, a cut-off value of serum HA levels ≥ 34.2 ng/mL (AUROC = 0.879) was determined for separating patients with significant liver fibrosis, METAVIR ≥ F2, and a cut-off value of ≥78.6 ng/mL (AUROC = 0.908) was used for the diagnosis of liver cirrhosis, F4 [17].

### 4.6. Statistical Analyses

The statistical analyses and graphing were performed using GraphPad Prism 5 for Windows (GraphPad Inc., La Jolla, CA, USA). The descriptive analysis is presented as median or mean ± standard error. We used D’Agostino-Pearson test to verify normal distribution. The analyses of comparative data were performed using Mann Whitney or *t* tests to verify the difference between healthy and HCV-infected blood donors, the Kruskal-Wallis test, the Friedman test to verify the results of the two ELISA experiments and the Spearman test to verify the correlations between liver biomarkers in the serum. A receiver operating characteristic (ROC) analysis was applied to determine the optimal HA or ALT levels for the identification of viremia, using HCV RNA testing as the standard. We calculated the sensitivity (the ability of the test to identify positive results), specificity (the ability of the test to identify negative results), positive predictive value (PPV) and negative predictive value (NPV) of the optimal S/CO ratio to predict HCV viremia. A 95% confidence interval of mean (CI_95%_) and a 5% level of significance were adopted; therefore, the results with a *p*-value less than or equal to 0.05 were considered significant.

## 5. Conclusions

A high level of HA (≥34.2 ng/mL) in association with ALT (≥40 U/L) in serum can provide a good clinical opportunity to detect HCV-infected asymptomatic persons that potentially require a liver biopsy confirmation and antiviral treatment to prevent the development of advanced liver fibrosis or cirrhosis.

## Figures and Tables

**Figure 1 molecules-26-03892-f001:**
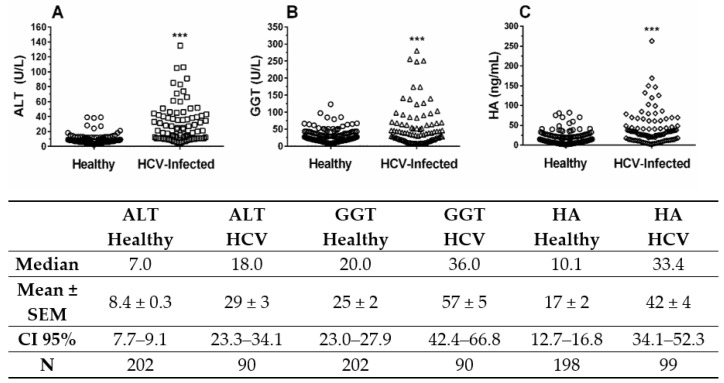
Hepatic enzymes and HA serum levels in function of HCV infection. (**A**) ALT serum level in heathy and HCV-infected blood donors. (**B**) GGT serum in heathy and HCV-infected blood donors (**C**) HA serum level in heathy and HCV-infected blood donors. Hepatic enzymes (U/L) and HA (ng/mL). Mann Whitney test, (***) Significant differences (*p* < 0.0001).

**Figure 2 molecules-26-03892-f002:**
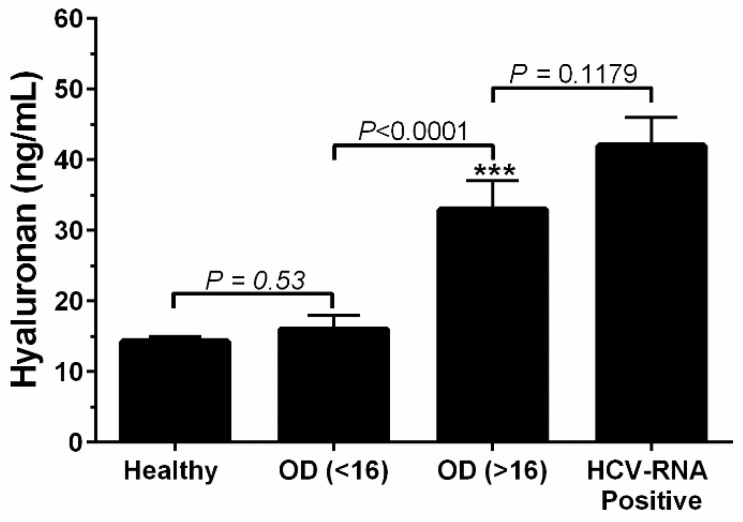
Hyaluronan as a function of the anti-HCV antibody serum level. Healthy blood donors: HA = 14.3 ± 0.7 ng/mL (N = 294); confirmatory anti-HCV ELISA: low antibody titer (OD < 16), HA = 16 ± 2 ng/mL (N = 43) and high antibody titer (OD ≥ 16), HA = 33 ± 4 ng/mL (N = 125); HCV-RNA positive blood donors: HA = 42 ± 4 ng/mL (N = 99). Optical Density (OD), HA (ng/mL) mean ± SEM; (***) significant differences by the Kruskal-Wallis test.

**Figure 3 molecules-26-03892-f003:**
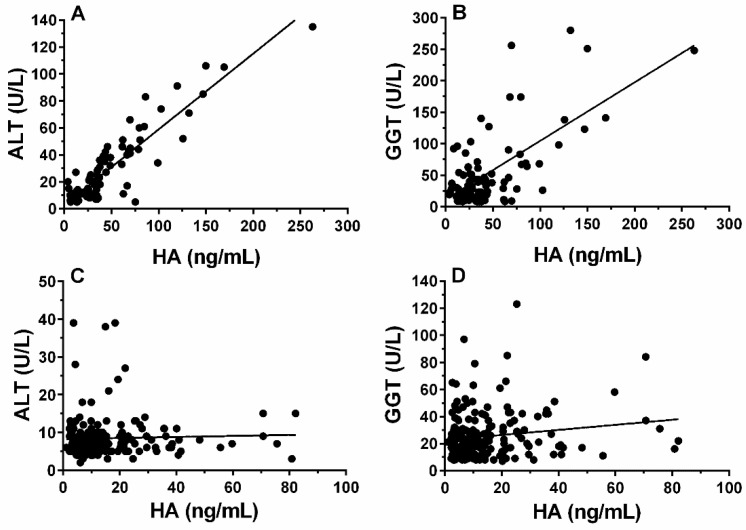
Correlation between hepatic enzymes and hyaluronan serum levels in HCV-infected and healthy blood donors. (**A**) ALT Spearman correlation with HA in HCV-infected blood donors (N = 87, ρ = 0.848, CI_95%_ = 0.773–0.899, *p* < 0.0001). (**B**) GGT Spearman correlation with HA in HCV-infected blood donors (N = 87, ρ = 0.442, CI_95_*_%_* = 0.249–0.601, *p* < 0.0001). (**C**) ALT Spearman correlation with HA in healthy blood donors (N = 194, ρ = 0.0618, CI_95%_ = −0.084–0.205, *p* = 0.3921). (**D**) GGT Spearman correlation with HA in healthy blood donors (N = 194, ρ = 0.1048, CI_95%_ = −0.041–0.246, *p* = 0.1468).

**Figure 4 molecules-26-03892-f004:**
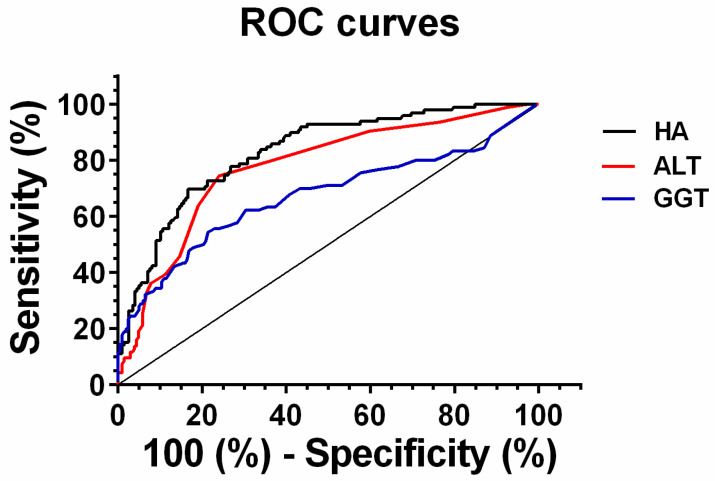
Receiver operator characteristic (ROC) curves. The relationship between the sensitivity (true positives) and specificity (false positives) for three diagnostic parameters, hyaluronic acid (**─**), ALT (─) and GGT (─), in determining HCV infection as compared to HCV-RNA as a predictor of HCV viremia. The levels of all three serum markers were determined in HCV-infected blood donors, and the data were used in the ROC statistical analysis to compare the diagnostic accuracy of the three parameters. The area under ROC curves (AUROC) were (Mean ± SE), GGT = 0.67 ± 0.04, ALT = 0.75 ± 0.03, and HA = 0.82 ± 0.02, respectively.

**Table 1 molecules-26-03892-t001:** Diagnostic performance of different cut-off points for the anti-HCV antibody titers as a predictor of HCV viremia.

Antibody Titers	HCV-RNA Positive	HCV-RNA Negative	
OD >16	96	29	PPV = 96/125 (76.8%)
OD ≤ 16	03	180	NPV = 180/183 (98.4%)
	Sensitivity = 96/99 (97.0%)	Specificity = 180/209 (86.1%)	Total = 308

**Table 2 molecules-26-03892-t002:** Diagnostic performance of the high HA serum level (HA ≥ 34.2 ng/mL) as a predictor of HCV viremia in blood donors with asymptomatic hepatic injury (ALT ≥ 40 U/L).

HA (ng/mL)	HCV-RNA Positive and ALT ≥ 40 U/L	HCV-RNA Negative and ALT < 40 U/L	
HA ≥ 34.2	24	20	PPV = 24/44 (55%)
HA < 34.2	01	184	NPV = 184/185 (99.5%)
	Sensitivity = 24/25 (96%)	Specificity = 184/204 (90%)	Total = 229

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
