# Peer review of "Diagnostic Accuracy of Serum Hyaluronan for Detecting HCV Infection and Liver Fibrosis in Asymptomatic Blood Donors"

_molecules, 2021, doi:10.3390/molecules26133892_

Round 1

Reviewer 1 Report

Major Comments

This paper is testing the diagnostic value of elevated hyaluronan levels in HCV positive blood donors.

I have one major problem with the message of this manuscript. The conclusions of the Abstract (line 28) and that of the Conclusion section (line 375) are different. Further, the last line of the Introduction section states a completely different, third conclusion, which confuses the reader right at start. Unfortunately, this confusion continues throughout the manuscript.

  1. Clear aims of the investigation should be stated in the Introduction.
  2. The authors should decide what they intend to focus on, and confusing notes should be avoided. I think the conclusion in line 375 would do. Then, the full manuscript should be restructured focusing on the primary aim.
  3. Nothing is mentioned, either in the Abstract or in the full manuscript, on the measurement technique of liver firbrosis and its degree (F0-F1-F2-F3). This is essential, as several comparisons with the severity of fibrosis are made in the Results section (lines 130, 133, 136 etc.). How was the presence/absence or severity of hepatic fibrosis measured for comparison with the HA data?

Minor notes

Figure 2, legend:

“Hepatic enzymes serum levels in function of HCV infection.”

Should read: Hepatic enzymes and hyaluronan serum levels in function…..

Hyaluronic acid and hyaluronan both are used throughout the manuscript; please choose one of them.

Table 2. Negtive predictive value 100%. This is by statistics only, in fact, 1 of the 185 pts is missed by the test. This, once data are shown, has to be discussed.

Discussion line 220:

“This finding did not differ from some studies in the literature, which HCV-RNA positive patients [33,34].”

‘which’ should read: with

English needs some edeiting.

Author Response

Response to Reviewer 1 Comments

MAJOR

Point 1: I have one major problem with the message of this manuscript. The conclusions of the Abstract (line 28) and that of the Conclusion section (line 375) are different. Further, the last line of the Introduction section states a completely different, third conclusion, which confuses the reader right at start. Unfortunately, this confusion continues throughout the manuscript.

Clear aims of the investigation should be stated in the Introduction.

 Response 1: We appreciate these observations and fully agree with them. We have insert at the end of the Introduction section of the Revised Manuscript, page 2, lines 80-85, the following sentence to address this issue: “Our main purpose here is not to create a method to test HCV-infected blood donors in-stead of HCV PCR, but rather to provide input to validate the serum HA detection to identify asymptomatic HCV-infected patients who are developing liver disease silently. Here, the present data give a possibility to non-invasively reveal liver fibrosis and initiate medical investigation and if needed treat blood donors found to be HCV RNA positive before they develop major liver disease.”

Also, the conclusions of the Abstract (line 28) and that of the Conclusion section (line 375) were corrected were corrected and standardized in the revised version of the Abstract: “A high level of HA (≥ 34.2 ng/mL) in association with ALT (≥ 40 U/L) in serum can provide a good clinical opportunity to detect HCV-infected asymptomatic persons that potentially require a liver biopsy confirmation and antiviral treatment to prevent the development of advanced liver fibrosis or cirrhosis”.

Point 2: The authors should decide what they intend to focus on, and confusing notes should be avoided. I think the conclusion in line 375 would do. Then, the full manuscript should be restructured focusing on the primary aim.

Response 2: We appreciate these observations and fully agree with them. The full manuscript was completely restructured focusing on the primary aim. “Our main purpose here is not to create a method to test HCV-infected blood donors in-stead of HCV PCR, but rather to provide input to validate the serum HA detection to identify asymptomatic HCV-infected patients who are developing liver disease silently. Here, the present data give a possibility to non-invasively reveal liver fibrosis and initiate medical investigation and if needed treat blood donors found to be HCV RNA positive before they develop major liver disease”.

Point 3: Nothing is mentioned, either in the Abstract or in the full manuscript, on the measurement technique of liver firbrosis and its degree (F0-F1-F2-F3). This is essential, as several comparisons with the severity of fibrosis are made in the Results section (lines 130, 133, 136 etc.). How was the presence/absence or severity of hepatic fibrosis measured for comparison with the HA data?

Response 3: We thanks these observations. We have inserted at the page 5, lines 153-157 of the revised version of the manuscript, the following sentence to address this issue: Previously, we demonstrated in a cohort of 206 biopsy patients with chronic hepatitis C that HA >34.2 ng/mL can predict, with high certainty (AUROC 88%), significant liver fibrosis (³F2) according to the METAVIR scoring system [17]. Therefore, the HA cut-off value greater that 34.2 ng/mL was chosen for identifying presence of significant liver fibrosis in blood donors.”

Also, we have inserted in the Materials and Methods, page 11, lines 384-381, the section 4.5.     Determination of Liver Fibrosis

Determination of liver fibrosis in blood donors was performed according to the level of serum HA concentration. Serum HA was the parameter that alone presented the best diagnostic accuracy in the assessment of hepatic fibrosis in a total of 206 hepatitis C virus RNA-positive biopsied patients [17]. HA levels strongly correlated with liver disease stage, a cut-off value of serum HA levels ³34.2 ng/mL (AUROC = 0.879) was determined for separating patients with significant liver fibrosis, METAVIR ³F2, and a cut-off value of ³78.6 ng/mL (AUROC = 0.908) was used for the diagnosis of liver cirrhosis, F4 [17].

Minor notes

Point 4: Figure 2, legend: “Hepatic enzymes serum levels in function of HCV infection.”

Should read: Hepatic enzymes and hyaluronan serum levels in function…

 Response 4: We appreciate these observations and fully agree with them. The legend of the new Fig. 2 at page 4, line 123 was corrected in the revised version of the manuscript as suggested: “Figure 2. Hepatic enzymes and HA serum levels in function of HCV infection”

 Point 5: Hyaluronic acid and hyaluronan both are used throughout the manuscript; please choose one of them.

Response 5: We chose to use the word hyaluronan in the revised version of the manuscript.

Point 6: Table 2. Negative predictive value 100%. This is by statistics only, in fact, 1 of the 185 pts is missed by the test. This, once data are shown, has to be discussed.

Response 6: In the Table 2 the NPV was corrected to 184/185 (99.5%) in the revised version. NPV is not intrinsic to the test, it depends also on the prevalence, our negative predictive value of 100% was corrected by the prevalence of HCV infection.

Point 7. Discussion line 220: “This finding did not differ from some studies in the literature, which HCV-RNA positive patients [33,34].”‘which’ should read: with

Response 7. Thank you for this correction. Please see this correction in Discussion section, page 9, line 254 of the revised manuscript: These results are similar to data found in the literature, with HCV-RNA positive patients [37,38]”.

Reviewer 2 Report

Review for the manuscript MOLECULES 1223294

Manuscript by Rodart IF et al “Diagnostic Accuracy of Serum Hyaluronan for Detecting HCV 2

Infection and Liver Fibrosis in Asymptomatic Blood Donors” describes possibility to efficiently and inexpensively diagnose HCV infection and liver fibrosis among blood donors by testing blood for the levels of hyaluronan (HA). Increased levels of HA appear to be more reliable marker of liver fibrosis than other liver enzymes, such as ALT.

Manuscript needs a revision before acceptance.

MAJOR

Firstly, one needs to state that this is not a method to test blood donors for HCV infection instead of HCV PCR. PCR test is 100% needed to detect and remove infected blood donations. HA test advances by Rodart IF et al gives a possibility to non-invasively reveal liver fibrosis and initiate medical investigation and if needed treat blood donors found to be HCV RNA+ before they develop major liver disease.

Authors need to add information about HA – metabolic pathway leading to HA and its function, to clarify why it emerges as a better marker of liver fibrosis (than ALT). This paragraph is instead placed at the very end of the discussion, and ends abruptly without concluding the main message of the manuscript. One starts with the role of HA, why HA levels increase in liver disease, and then comes to possibility to diagnose liver disease, namely fibrosis, by measuring HA levels.

Figure 1 – no need to demonstrate results of HCV genotyping by showing agarose gels with PCR products, figure panels A and B could be given as supplementary.

Section 2.2 – needs to be shortened as it repeats information in Figure 2. Figure 2 can instead be given with data table beneath showing values and differences.

Lines 87-88 contradicts the rest of the paragraph, statements above and below. Needs to be transferred at the end of the paragraph, saying “However ALT levels were elevated in only 25%,a nd GGT, in 29%” (indicating that one needs an additional test to reveal cases of liver disease due to HCV infection).

Figure 3 – please, show all pair-wise statistical comparisons in the graph, instead of listing them in the text. Instead of “HCV positive”, use term HCV RNA positive”.

Figure 4 – show also panels illustrating absence of correlations between ALT, GGT and HA in healthy donors.

Lines 211-213 – of no relevance to the topic of this manuscript. This section could be concluded by the statement that HA test can complement testing of blood banks for HCV using anti-HCV antibody tests, as it can reveal viremic donors negative for anti-HCV antibodies. However, it needs to be clearly stated that HCV RNA test needs to be done for blood banks as the only reliable.

MINOR

In the abstract, “detect significant liver fibrosis…before they develop major liver fibrosis” – what is significant and what is major, those are not the terms to use in diagnostics, or in manuscript devoted to diagnostics.

Page 2  line 53 – particular liver disease, what is meant by this term?

Line 58 – liver biomarker not optimal – biomarker of liver function.

Line 63 – OD>16 – unclear, what is OD as an abbreviation, how/based on what it was introduced.

Line 157 – decipher what is ROC curve first time this term is used in the section 2.5

Line 204 – “our data corroborate this finding” – the data do not corroborate findings, but complement them identifying cases of HCV infection characterized by HCV RNA+ test and low levels of anti-HCV antibodies (OD<16).

Line 209-210 - There is no contrast between 0.38-2.6% and 0.2% on one side, and 1.6% on the other side. Specifically, 1.6% is within 0.38-2.6% interval.

Line 220-221 – please, correct language “This finding did not differ from some studies in the literature”  Finding should be compared to findings, “some studies” is no good either, it indicates that the other studies may have found the opposite.

Line 255-257 – language “HA level… associated with … and probable also indentified…” – better say “could potentially identify”.

Line 262 – need to add “anti-HCV” - “high anti-HCV antibody titers”.

Round 2

Reviewer 1 Report

All relevant issues appropriately addressed.

Author Response

Comments and Suggestions for Authors:

All relevant issues appropriately addressed.

Response:  I would like to thank Reviewer 1 for all their suggestions and criticisms that allowed for the improvement of this manuscript.

Reviewer 2 Report

Authors have modified the manuscript taking care of all critical elements. One still remains - in 2021, one should not illustrate products of RT-PCR confirming HCV genotypes. These tests are run for >20 years, there are test systems available, commercial as well as in-house. Even studies done in low-resource settings do not illustrate PCR products - see for example https://virologyj.biomedcentral.com/articles/10.1186/s12985-019-1214-9 With this, I strongly recommend to the authors to place photos of the agarose gels with PRC products (Figure 1. HCV genotyping of blood donors) in the supplement. The manuscript is fine and can be accepted after this correction.  

Author Response

Comments and Suggestions for Authors:

Authors have modified the manuscript taking care of all critical elements. One still remains - in 2021, one should not illustrate products of RT-PCR confirming HCV genotypes. These tests are run for >20 years, there are test systems available, commercial as well as in-house. Even studies done in low-resource settings do not illustrate PCR products - see for example https://virologyj.biomedcentral.com/articles/10.1186/s12985-019-1214-9. With this, I strongly recommend to the authors to place photos of the agarose gels with PRC products (Figure 1. HCV genotyping of blood donors) in the supplement. The manuscript is fine and can be accepted after this correction.

Response: As recommended by Reviewer 2, the "Figure 1. HCV genotyping of blood donors..." was placed in the Supplementary Materials. Also, I would like to thank Reviewer 2 for all the suggestions and criticisms made that allowed for the improvement of this manuscript.